# Greenhouse Gas Emissions and Crossbred Cow Milk Production in a Silvopastoral System in Tropical Mexico

**DOI:** 10.3390/ani13121941

**Published:** 2023-06-09

**Authors:** Lucero Sarabia-Salgado, Bruno J. R. Alves, Robert Boddey, Segundo Urquiaga, Francisco Galindo, Gustavo Flores-Coello, Camila Almeida dos Santos, Rafael Jiménez-Ocampo, Juan Ku-Vera, Francisco Solorio-Sánchez

**Affiliations:** 1Department of Ethology, Wildlife and Laboratory Animals, Faculty of Veterinary Medicine and Animal Science, National Autonomous University of Mexico (UNAM), Ciudad Universitaria, Mexico City C.P. 04510, Mexico; lucy_34_88_sarsal@hotmail.com (L.S.-S.); galindof@unam.mx (F.G.); gfc15sheep@gmail.com (G.F.-C.); 2EMBRAPA/Agrobiologia, Brazilian Corporation for Agricultural Research—National Centre for Agrobiology Research, Seropédica 23891-000, RJ, Brazil; bruno.alves@embrapa.br (B.J.R.A.); robert.boddey@embrapa.br (R.B.); segundo.urquiaga@embrapa.br (S.U.); 3Department of Soil Sciences, Federal Rural University of Rio de Janeiro (UFRRJ), Seropédica 23890-000, RJ, Brazil; milaema04@gmail.com; 4National Institute for Forestry, Agriculture and Livestock Research—INIFAP, Experimental Field Valle del Guadiana, Durango C.P. 34170, Mexico; jimenez.rafael@inifap.gob.mx; 5Animal Nutrition Department, Campus of Animal Production and Biological Sciences, Autonomous University of Yucatán, Merida C.P. 97000, Mexico; kvera@correo.uady.mx

**Keywords:** nutrient cycling, tropical grassland, urine, N balance, milk production

## Abstract

**Simple Summary:**

Currently there is an urgent need to modify food production systems, including the influence of ruminants, due to extensive land use and environmental impacts. Grazing cattle excreta emit considerable amounts of methane and nitrous oxide. The objectives of this work were to assess the production and quality of the forage, milk production, and methane and nitrous oxide emissions from the cattle feces and urine in two production systems: conventional grazing (grass in a monoculture) and a silvopastoral system (association of leguminous shrubs with grass). The inclusion of legumes in the diet of grazing cattle increases forage quality and reduces the methane and nitrous oxide emissions from urine and feces.

**Abstract:**

In Mexico, pasture degradation is associated with extensive pastures; additionally, under these conditions, livestock activities contribute considerably to greenhouse gas (GHG) emissions. Among the options to improve grazing systems and reduce GHG emissions, silvopastoral systems (SPS) have been recommended. The objectives of this work were to quantify the N outflow in a soil–plant–animal interface, as well as the CH_4_ emissions and milk production in an SPS with woody legumes (*Leucaena leucocephala*) that is associated with stargrass (*Cynodon nlemfuensis*). This was then compared with stargrass in a monoculture system (MS) in the seasons (dry and rainy period) over a two-year period. Dung was collected from the animals of each of the grazing systems and applied fresh to the land plots. Fresh dung and urine were collected from the cows of each grazing system and were applied to the experimental plots. In addition, the soil CH_4_ and N_2_O contents were measured to quantify the emissions. Average milk yield by seasons was similar: MS (7.1 kg per animal unit (AU)/day^−1^) and SPS (6.31 kg per AU/day^−1^). Cows in the MS had a mean N intake of 171.9 g/UA day^−1^ without seasonal variation, while the SPS animals’ mean N intake was 215.7 g/UA day^−1^ for both seasons. For the urine applied to soil, the N_2_O outflow was higher in the MS (peak value = 1623.9 μg N-N_2_O m^−2^ h^−1^). The peak value for the SPS was 755.9 μg of N-N_2_O m^−2^ h^−1^. The N_2_O emissions were higher in the rainy season (which promotes denitrification). The values for the feces treatment were 0.05% (MS) and 0.01% (SPS). The urine treatment values were 0.52% (MS) and 0.17% (SPS). The emissions of CH_4_ showed that the feces of the SPS systems resulted in a higher accumulation of gas in the rainy season (29.8 g C ha^−1^), followed by the feces of the MS system in the dry season (26.0 g C ha^−1^). Legumes in the SPS helped to maintain milk production, and the N_2_O emissions were lower than those produced by the MS (where the pastures were fertilized with N).

## 1. Introduction

The management of cattle production systems is highly variable and can show different levels of intensity regarding the use of pastures. In Mexico, as in most tropical countries, extensive traditional cattle production is characterized by being highly extractive, with low production levels and high emissions of greenhouse gases [1]. These systems are practiced in more than 80% of the land devoted to animal production in Mexico, the largest proportion being extensive grazing areas (between 80 and 120 million hectares), with an approximate cattle herd of 31.9 million head [2], resulting in low stocking rates and low productivity.

Due to pasture degradation and low carrying capacities, cattle ranchers frequently implement additional strategies of supplementary feeding, which increase production costs. Alternatively, they seek new areas for forage cultivation, thus accelerating deforestation, loss of biodiversity, as well as generating livestock systems that are more vulnerable to climate change and which produce higher greenhouse gas (GHG) emissions. Pasture degradation is associated with the N cycle in the production system. Depending on the management adopted, the soil–plant–animal system results in a negative balance of N (losses of N > input), causing the inefficient cycling of N by cattle excreta that reduce pasture production over time and also contribute to its degradation [3]. Tropical cattle depend, to a large extent, on the production of grass. However, generally in tropical conditions, the soil has a low nitrogen content, which limits animal productivity. Most tropical grasses have high NDF and low CP content, resulting in a limited fermentation of DM, the long retention time of digesta, and the low absorption of volatile fatty acids from the rumen [4]. This leads to modest daily weight gains in growing cattle and considerable emissions of enteric methane. Additionally, the combination of the marked seasonality of the rains and poor soils considerably affects the availability of biomass, as well as the quality and digestibility throughout the year.

In general, there are two ways to improve the quality and productivity of forages. One strategy is through the application of nitrogenous fertilizers. The other, is through the incorporation of nitrogen-fixing legumes [5]. Additionally, legumes are one of the best sources of forage with a high protein content for animals, and they have the ability to contribute to soil fertility. An increase in the production of biomass and quality has a positive impact on the increase in the number of animals, weight gain, and milk production.

Additionally, the CH_4_ emissions per unit of product could be considerably reduced [6,7]. On the other hand, more productive and healthy pastures allow a greater accumulation of C in the soil, which contribute to further mitigation of GHG emissions [8].

Increasing the amount of N circulating in the system can increase N_2_O emissions [9] and, theoretically, reduce or neutralize the gains that are obtained with the reduction in other emissions [10]. The use of synthetic nitrogen fertilizers is of limited efficiency since, in global terms, only 50% of the N applied is used by plants; the rest is immobilized in the soil and/or lost to the atmosphere [11]. Regarding the impact on greenhouse gas emissions, nitrogen fertilization with 100 kg of N ha^−1^ can result in emissions of approximately 1 mg of CO_2_eq ha^−1^, which require fossil energy for ammonia synthesis, processing, and transportation [12], as well as direct and indirect N_2_O emissions after contact with the ground [13]. On the other hand, with the use of legumes, the energy to biologically fix N_2_ comes from photosynthesis; moreover, this also promotes the mitigation of GHG emissions [14]. Although there are different studies on the benefits of silvopastoral systems regarding productivity and animal welfare [15], limited research has been carried out to evaluate their effect on CH_4_ and N_2_O emissions and nitrogen balance [16], including comprehensive productive aspects such as milk production [7].

The use of Leucaena (*Leucaena leucocephala* (Lam.)) de Wit. in a system of rows in pastures is a practice that has been spreading in Mexico for the purpose of increasing forage quality at a lower cost [17,18]. This legume species is highly palatable, and it can accumulate large amounts of biologically fixed N_2_ (between 100 and 300 kg of N ha^−^^1^ year^−^^1^ [19]). However, the tannin content in the leaves can limit the use of N by animals [18,20]. Consumption of Leucaena can stimulate voluntary intake by animals due to a higher crude protein intake since the digestibility and use of N depends on the energy intake sources. The presence of tannins can also modify digestibility, and the protein complexes formed in the rumen can result in greater N excretion in the feces [21]. On the other hand, the lower excretion of N in the urine reduces the potential for N loss due to ammonia volatilization and N_2_O emissions [22], thus maintaining the nutrients in the system. This highlights the considerable productive and environmental potential of production systems that use legumes, as is mentioned by Jenzen et al. [23]. More studies that consider the diversity of species and possible management systems are needed to consolidate the benefits of this practice. The objective of this study was to quantify, in two production systems, the supply of forage, milk production throughout the year, feed consumption, the N content of feces and urine, as well as to estimate the emissions of the N_2_O and CH_4_ derived from the deposition of feces and urine in the soil.

## 2. Materials and Methods

### 2.1. Milk Production and Nitrogen Excretion

#### 2.1.1. Site Description

This study was conducted at the Campus of Agricultural and Biological Sciences of the University of Yucatan—which is located in the city of Merida, Yucatan State, Mexico—from May to August 2017 and from November 2017 to February 2018. The average temperature was 29.2 °C from May to September 2017 and 24.6 °C from November to March 2018. The experimental site is located at 21°15′ N and 83°32′ W, with an altitude of 10 m.a.s.l. The climate of the region is hot and humid, with rain in the summer [24]. May–September 2017 recorded an average temperature of 29.2 °C. The period of November 2017–March 2018 recorded a temperature of 24.6 °C on average. The soil of the area is Rendzic Leptosol [25]. To reduce experimental error (soil variability) in the soil characterization, soil samples were taken in a randomized block (Table 1).

Table 1 shows the physical and chemical properties of the soil at the beginning of the experiment. The information on the climate conditions for the months of the experiment were collected at the Meteorological Station of the Experimental Dairy Cattle Unit, approximately 200 m from the experimental area.

Evaluations were performed in the SPS and the MS. The SPS associated *C. nlemfuensis* (Vanderyst cv. Tifton 85) and *L. leucocephala* (Lam. de Wit cv. Cunningham). The density of Leucaena plants was 27,000/ha, which were distributed by being separated by 2 m within rows, and the space between the plants was 40 cm. Between rows, *Cynodon nlemfuensis* was established. The study area was subdivided into 28 paddocks of 0.4 ha (14 paddocks in each system). Twelve multiparous lactating crossbred (Holstein and Brown Swiss × Zebu) cows at a 506 kg live-weight were used. Based on their milk yield, the cows were divided into homogenous groups that comprised 6 animals. One group received the experimental treatment of the SPS, and the other group received the MS treatment. A control group (*n* = 6 cows) grazed in the traditional monoculture system (MS). Two months before beginning the experiment, the *C. nlemfuensis* grass was fertilized with 100 kg of N ha^−1^, which was split into three doses (approximately 70 kg of urea ha^−1^). In summer, each of the paddock cows grazed for 3 days before moving on the next grazing paddock, which was then left unoccupied for 27 days (rotational grazing), or until their complete recovery. In the rainy season, the stocking rate was 4.8 AU ha^−1^, and in the dry season 3.4 AU ha^−1^. Both groups grazed from 13:00 to 06:00 h the next day. The paddocks were irrigated to maintain forage yield. In the two kinds of treatments (SPS, MS), each cow was supplemented daily with 3 kg of concentrate (dry basis), which was composed of 63% sorghum grain, 27% soy hulls, 8% soybean, 1% calcium, and 1% mineral salts, totaling 14% of the crude protein and digestibility of 80% in the final mixture.

Forage biomass was quantified just before the animals entered the paddock and after the animals left. Forage biomass in the MS group was quantified using a 0.5 m × 0.5 m (0.25 m^2^) metal quadrangle, which was a modification of Cox’s technique [26]. Each time, ten grass samples were taken in a zig-zag pattern from each paddock, and the grass was cut inside a quadrangle that was 5 cm above the ground. Then, the grass was weighed. The SPS forage availability was recorded following Bacab-Pérez and Solorio-Sánchez [27]: the leaves and young stem of the Leucaena (the edible forage) inside a 4 m^2^ quadrangle were harvested. The grass inside the quadrangle was also cut 5 cm from the ground. On all occasions, grass samples were taken from each paddock. A selection of subsamples of grass and Leucaena were oven-dried at 60 °C to a constant weight [28]. The Kjeldahl method [29] was used for N content, the filter bag technique [30] for the acid detergent fiber (ADF), and the neutral detergent fiber (NDF) was determined in accordance with the method described by Van Soest et al. [31]. Simultaneously, an experiment to assess the CH_4_ and N_2_O emissions from the excreta was also carried out.

#### 2.1.2. Milk Production

The cows were mechanically milked once a day at 7:00 h, and milk production was carried out by weighing the milk during each period (with the presence of the calf). In the last 6 days of each period, the crude protein concentration was determined in 100 mL of the milk samples, which were obtained at milking and were analyzed the same day using the Lactoscan ultrasonic milk analyzer (Milkotronik Ltd., Nova Zagora, Bulgaria). The formula used to convert the protein into N was the following:(1)$$N=\frac{CP}{6.25}$$

#### 2.1.3. Estimation of Forage Intake

The dry matter intake (kg DM day^−1^) was assessed using chromic oxide (Cr_2_O_3_) as described by Pond et al., 1989 [32].

Dry matter intake (voluntary intake) was calculated using this equation:(2)Voluntary Intake ((gDM)/day)=Fecal production ((gDM)/day)[1−(Digestibility/100)]

For the estimation of the N consumed by the animals, forage samples were collected to quantify the N content with the Kjeldahl method [29]. For validation, the feed intake was also estimated based on forage digestibility and fecal production according to this equation:(3)$$Intake=\frac{Fecal\;production}{\left(1-digestibility\right)}$$

Fecal production was quantified with the external chromium oxide marker technique [32]. To evaluate the composition of the forage intake, samples of the plants that simulated the grazing of the animals were taken and were, in each period, evaluated [32]. In the SPS treatment, separate grass and legume samples were collected. After the collection, samples were identified, separated into stem and leaves, and taken to the laboratory. Samples were then oven-dried at 60 °C until at a constant weight [28] and grounded in a Wiley-type mill, with a 1 mm mesh sieve. Forage samples were analyzed, via the Kjeldahl method [29], for N content and for the in vitro digestibility of the consumed material [31]. The proportion of the legumes in the consumed forage was determined from the results of the ^13^C abundance of feces and the digestibility of grass and Leucaena, as described by Macedo et al. [33]. N intake was estimated based on the total N analysis of the composite grass, and the legume samples were obtained by the simulated grazing.

#### 2.1.4. Fecal and Urinary Production

Fecal production was estimated using the chromium oxide indicator [34]. The indicator was supplied for 16 days (10 g/d) according to Macedo et al. [33] (10 days of adaptation and six days of feces collection). While milking, approximately 200 g of feces samples were collected from the animals’ rectums.

The samples were identified and dried in a forced air circulation oven at 60 °C for 72 h. After pre-drying, samples from the same animals and corresponding treatment were milled (1 mm) and analyzed for the concentration of Cr_2_O_3_ via atomic absorption spectrophotometry according to Fenton and Fenton [35]. Fecal production (FP) was calculated by this formula:(4)$$FP(kg)=(\frac{Mf}{Mm})\times 10$$
where Mf and Mm are, respectively, the dry stool (M) and marker masses in the sample. The marker mass was administered daily.

Samples were also analyzed for the total content of N and for the abundance of ^13^C in order to determine the proportion of C in the feces derived from the legume. This was performed according to the following formula:(5)Proportion of C in the feces(%)=[(Cf13−Cg13)÷(Cl13−Cg13)]×100
where ^13^Cf, ^13^Cg, and ^13^Cl are, respectively, the natural ^13^C abundance of feces, grasses, and legumes.

The urine and fecal samples were taken at the same time. Urine volume and the concentration of N in the urine were estimated using “spot” urine collection [36]. Approximately 30 mL of urine was collected from each animal, then mixed with 6 mL of 20% sulfuric acid to stabilize the medium. The nitrogen concentration was determined from each sample via the Kjeldhal method, and creatinine by the colorimetric method (a total excretion of 27 mg of animal creatinine day^−1^ was assumed for the calculation of the urinary volume). The N excreted via urine was obtained by multiplying the urine volume by the concentration of N in the urine.

### 2.2. CH_4_ and N_2_O Emissions

#### 2.2.1. Experimental Design and Excreta Handling

An experimental grazing simulation was performed in a *C. nlemfuensis* grass paddock with a height of 15 cm. Five treatment plots of 1.5 m × 1.5 m were delimited as follows: (1) a control plot without excreta; (2) a MS plot where cattle urine was added; (3) a MS plot where cattle dung was added; (4) a SPS plot where cattle urine was added; and (5) a SPS where cattle dung was added. The design of the experiment was a randomized complete block, and each treatment had six replicates (30 plots). The rainy and dry seasons had a factorial design. The base of a 40 × 60 cm static chamber was inserted in the soil at the center of each plot to an average depth of 15 cm.

The urine and dung were freshly collected from the same cows and systems used in the first experiment. Approximately 15 kg of fresh dung was well mixed in a container until visually homogeneous. Upon collection, the dung from the six cows of the MS was combined, homogenized, and sampled to quantify the total N content. The same procedure was followed with the dung from the six animals in the SPS. Urine was processed the same way. The quantification of volatile solid content was performed through combustion at 550 °C in a muffle furnace.

To measure the N_2_O and CH_4_ contents, 2.0 kg of fresh dung was placed with the aid of a plastic ring (approximately 24 cm in diameter and 5 cm in height) in each plot at the center of the rectangular metal frame bottom of the static chamber, and care was taken to cover the area homogeneously. To simulate the urination of the animal, 1.2 L of the fresh combined sample of urine was applied per chamber, ensuring the entire area delimited by the rectangular metal frame (0.24 m^2^) was moistened. The amount of dung and urine applied in each plot corresponded to a single excretion event of an adult animal: 1.5–2.7 kg of dung [37] and 0.8–1.7 L of urine [38].

The evaluations during the rainy season began on 20 May 2017 and ended on 20 August 2017. During dry season evaluations, the samplings began on 20 November 2017 and ended on 20 February 2018. The procedures to collect and add dung and urine were the same in both seasons, but the position of the chamber within each plot was changed so that there was no overlap in the excreta applied in each period.

#### 2.2.2. Quantification of N_2_O and CH_4_ Emissions

To monitor the N_2_O and CH_4_, manually closed static top–bottom type chambers—similar to those described by Alves et al. [9]—were used (See Figure 1 below). The bottom, a rectangular iron frame 40 cm wide, 60 cm long, 15 cm high, was inserted into the soil. At the upper perimeter, a trough with dimensions of 2 cm × 2 cm (W and H) had a water-sealed connection with the top part. The latter had the same dimensions as the base, but its height was 25 cm when coupled to the base. An aluminized thermal insulation mantle minimized the temperature increase after it was put into place. Chamber bases were inserted into the soil up to the level of the trough one week before the beginning of the gas flux measurements; it remained in place until the end of the study to avoid interferences due to soil disturbance. The headspace of the chamber was always sampled between 09:00 and 11:00 as it was assumed that the GHG flux at this time represented the average of the fluxes of the day [9]. At the time of gas collection, the internal temperature of each chamber was measured with a digital thermometer to correct the gas fluxes. Air samples were collected with 60 mL polyethylene syringes at 0, 20, 40, and 60 min after placing the chamber. A volume of 10 mL was flushed out from the chamber and transferred to 20 mL chromatography vials within an hour after chamber sampling. Vials were prepared using an electric vacuum pump.

Gas sampling started two days before adding excreta to the plots (days “−2” and “−1”) during the rainy season, and this continued for 10 consecutive days after adding dung or urine. Subsequently, gas sampling was performed every two days for two weeks, and then once a week for a period of approximately three months. If it rained, additional sampling was carried out for two or three consecutive days. In the dry season, gas sampling also started two days before placing the excreta, but continued for five consecutive days after excreta was added, followed by weekly samplings for approximately three months (at which time the dry season ended). The N_2_O and CH_4_ concentrations in the gas samples were analyzed simultaneously in the Nutrient Cycling Laboratory of EMBRAPA Agrobiology (Brazil) with a gas chromatograph Shimadzu GC 2014. The device was equipped with a flame ionization detector (FID) for CH_4_ and an electron capture detector (ECD) for N_2_O. The fluxes were calculated with the following equation:(6)$$f=(\frac{∆C}{∆t})\times (\frac{M}{Vm})\times (\frac{V}{A})$$
where f is the gas flow; ΔC/Δt is the variation of the gas concentration over time; M is the molecular weight of the gas; Vm is the molecular volume at the sampling temperature; and V and A are, respectively, the chamber volume and the floor area covered by the chamber. After calculating the flows, the emissions for the monitoring period adopted in the experiment were estimated with the Newton–Cotes numerical integration technique, using the rectangle method.

#### 2.2.3. Emission Factors

The total emissions of N_2_O and CH_4_, which were integrated in the monitoring period, were computed for the areas with excreta (Ee), and for the control area without excreta (Ec). For excreta treatments, the emissions from the control area were divided. The total liquid of each treatment was divided by the total of N or C present in the corresponding excreta (Qe) to calculate the emission factor (EF):$$EF=\frac{\left(Ee-Ec\right)}{Qe}$$

### 2.3. Statistical Analysis

The variables measured in the first experiment of the grazing systems in rotational paddocks were analyzed by a mixed linear model. The categorical variables “grazing system” (MS and SPS) and “time of year” (rain and dry) were considered as a fixed effect and the field plot was considered as a random effect. The mixed linear model was processed using the lem4 package of the R software for analysis of the variance. The residuals from the mixed model were validated, by the Shapiro–Wilk test, via a verification of the normality of the errors, and the homogeneity was verified by the Bartlett test. The separation of means was performed by the pairwise Tukey test at a 5% probability. In the second experiment, the N_2_O and CH_4_ emissions from excreta was subjected to ANOVA, under the assumptions of normal distribution; mean values were separated based on the minimum significant difference by the Tukey test at 5% probability. The evaluated variables showed the results with the mean and the standard error of the mean.

## 3. Results

### 3.1. Weather Data

From the beginning of May to the end of August 2017, an accumulated precipitation of 403 mm was recorded with relatively well distributed rains, and with the average daily maximum and minimum temperatures of 35.7 and 22.7 °C, respectively (Figure 2A). In the period from November 2017 to March 2018, the volume of rainfall was only 75 mm, with average maximum temperatures of 30.5 °C and minimum temperatures of 18.7 °C (Figure 2B).

The conditions observed in the middle of 2017 represent the summer climate of the region, with more frequent rains and higher temperatures in the rainy season. In the following period, low intensity and infrequent rains characterize the dry period, which also includes slightly lower temperatures that were below 20 °C for several days.

### 3.2. Influence of Diet on Milk Production

The forage in the MS and SPS systems in the rainy season was 7.3 ± 0.9 and 6.9 ± 0.7 kg of milk AU^−1^ day^−1^, respectively (Figure 3). In the dry season, milk production decreased slightly to 6.9 ± 1.0 and 5.7 ± 0.8 kg AU^−1^ day^−1^ for the MS and SPS, respectively. Despite the apparent reduction, there was no statistical difference between the systems at each season of the year and between the seasons of the year for each system. As can be seen in Figure 3, only a slight trend of lower milk production was observed in the SPS system.

### 3.3. Forage Intake and N Content, and the Proportion of Legumes in the Forage Consumed

With the presence of livestock, N recycling in the grasslands can be modified depending on what the animals consume. Forage intake (Figure 4) was estimated using the standard technique that starts with the quantification of fecal production that uses the external marker of chromium oxide combined with the information on forage digestibility. Feces represent the indigestible fraction of the forage.

In the SPS system, the numbers were similar, with 1.7 ± 0.2 kg of DM U.A.^−1^ day^−1^ and 1.5 ± 0.3 kg of DM U.A.^−1^ day^−1^ for the rainy and dry seasons, respectively. Forage digestibility was 65% for stargrass and 76% for Leucaena in the rainy season, and this decreased to 52% and 69% in the dry season, respectively.

The intake of Leucaena in the SPS was 0.97 ± 0.5 and 0.84 ± 0.3 kg of DM AU^−1^ day^−1^ for the rainy and dry season, respectively (Figure 4).

### 3.4. Quantification of Urinary Production, Fecal Production, and Nitrogen Balance

The urine volume for the MS were 13.5 and 13.2 L of AU^−1^ ha^−1^; furthermore, for the SPS, it was 11.3 and 13.2 L AU^−1^ ha^−1^ for the rainy and dry seasons, respectively. There was no statistical difference between the systems and times of the year (Table 2). In the MS, fecal production was 1.55 kg of DM AU^−1^ day^−1^ and 1.7 kg of DM AU^−1^ day^−1^ for the rainy and dry season, respectively. In the SPS system, the numbers were similar, with 1.75 kg of DM AU^−1^ day^−1^ and 1.53 kg of DM AU^−1^ day^−1^ for the rainy and dry seasons, respectively.

In the rainy season, the N intake was lower in the MS, totaling 115 g of N AU^−1^ day^−1^, including the N present in the concentrate that was offered daily (Table 3). In the SPS system, the presence of Leucaena increased the N intake, which reached 172 g of N AU^−1^ day^−1^. In the dry season, the N intake was reduced in relation to the rainy season, but there were no differences between the systems (being a little above 142 g of N AU^−1^ day^−1^, on average).

### 3.5. Nitrous Oxide and Methane Emissions from Bovine Excreta

The highest daily flows of N_2_O occurred in the first 6 days of collection for the two systems evaluated (MS and SPS); after that period, the flows were rather low. The treatment of urine from the MS presented a higher emission of N_2_O, with a peak of approximately 1624 µg of N-N_2_O m^−2^ h^−1^ for the rainy season and 472 µg of N-N_2_O m^−2^ h^−1^ for the dry season, respectively. The N_2_O emission value in the MS stool treatment was 566 µg of N-N_2_O m^−2^ h^−1^ for the rainy season and 126 µg of N-N_2_O m^−2^ h^−1^ for the dry season—which were slightly higher than the control values. In the SPS system, the urine treatment flows were higher compared to the feces and the control, and the highest peaks were observed in the excreta of the MS.

The daily fluxes of CH_4_ were also present in the different treatments throughout the days of the experiment (Figure 5). Urine emissions were practically zero for both of the systems in the rainy and dry seasons, as well as for the control group. The highest flows occurred in the first three days for the treatment of feces for both management systems (MS and SPS). Treatment with animal feces from the SPS system presented a higher emission peak (7.6 and 5.0 µg C-CH_4_ m^−^^2^ h^−^^1^) for rainy and dry conditions when compared to the treatment of feces originating from the animals in the MS system (4.8 and 7.0 µg of C-CH_4_ m^−^^2^ h^−^^1^ for the rainy and dry conditions, respectively).

When integrating the flows for the monitoring period, the highest emissions of N_2_O were observed with the addition of urine (119.1 and 80.1 g of N ha^−1^ h^−^^1^ for MS and SPS, respectively). The N_2_O fluxes after the addition of feces were low for both treatments, varying from 41.2 to 30.4 g of N ha^−^^1^ h^−^^1^.

The feces from the SPS systems resulted in a higher accumulation of gas (CH_4_) in the rainy season (29.8 g of C ha^−^^1^), followed by the feces from the MS system in the dry season (26.0 g of C ha^−^^1^).

### 3.6. Emission Factors

Treatments with feces presented N_2_O emission factors of 0.05 and 0.01% for pastures in the MS and SPS in the rainy season, respectively, while the respective emission factors for urine were 0.52% and 0.17% of the total N applied (See Table 4 below). For the dry season, the emission factors were lower, varying from 0.02 to <0.01% for the feces in the MS and SPS systems, respectively. For urine, the factors were higher compared to feces, although lower than those observed in the rainy season (0.05% in both grazing systems). Urine from the SPS system in the dry season had a higher N content than urine from the MS; however, the emission factor was the same, despite the reason not being clear.

## 4. Discussion

### 4.1. Influence of Diet on Milk Production

Despite having different diets (MS and SPS), there were small differences. In a previous study in the same experimental station and with cows of the same breed, Tinoco-Magaña et al. [39] reported a milk production of 10.6 kg per animal day^−^^1^. The animals had 4-h access to the SPS system and were offered 1 kg of sorghum grain as an energy supplement. Other studies reported a higher milk production than between 29 and 31 kg, which were found in the present study, but these were with the Leucaena being offered in a protein bank. Razz et al. [40] obtained 9.6 kg of animal milk day^−^^1^, and Faría et al. [41] found a production of 10.8 kg of animal milk day^−^^1^. A situation similar to that of the present study was verified by Garcia and Sanchez [42], who found similar milk production between cows (9.1 kg animal day^−^^1^), both with and without access to Leucaena (9.2 kg animal day^−^^1^). This was attributed to the fact that the pasture only covered the nutritional requirements of the cows. However, we cannot dismiss the possibility that there was an excess in the consumption of N that was derived or provided in the foliage of Leucaena.

Milk production based on tropical forages is generally limited due to the intake of low-quality forage, which often does not meet the requirements of grazing animals. Under these conditions, there is a large variation in milk production, ranging from 5 to 14 kg of animal day^−^^1^ [43]. During the dry season, animals are even more affected due to the reduction in the quality and supply of forage, especially in pastures. In these circumstances, the use of protein supplements is recommended in order to meet the cow’s maintenance and milk production requirements [44]. According to the NRC [45], a cow producing between 7 and 9 kg day^−^^1^ of milk requires a forage with approximately 11 to 12% CP. In this case, the use of legumes rich in N, such as Leucaena, is a good strategy to use in combination with low-quality forages. However, stargrass already has CP levels within this range, except in the MS during the rainy season. In this case, the supplementation of concentrates with 14% CP must have met the needs of the animals used in the study. This further supports the hypothesis that in both systems there was a high supply of crude protein, especially in the SPS where Leucaena contributed to an increase in the amount of N in the diet.

Additionally, the condensed tannins present in Leucaena occasionally positively influence animal production by forming complexes that increase the rate of protein outflow from the rumen, thus reducing its degradation and increasing animal productivity [46,47]. On the other hand, negative effects can occur since it is possible that these complexes are very stable and end up being excreted in the feces, reducing milk productivity and weight gain, and thus highlighting that the N excreted in these complexes does not necessarily originate in the diet but are due to physiological disorders caused by the intake of tannins [48].

In this study, the Leucaena contained 4.12% tannins in the rainy season and 1.09% tannins in the dry season. Surprisingly, high tannin contents in Leucaena in the summer period (3.0%), which rose to 3.8% in autumn and decreased in winter (0.5%), were also found by Foroughbakhch et al. [49] when evaluating 20 tree and shrub species in northeastern Mexico. Although tannins are considered anti-nutritional factors, according to Morales and Ungerfeld [50], the effects of tannins in ruminants feeding depend on several factors; among them, the animal species, composition of the diet, type of tannin (structure, molecular weight), and quantity [51]. The relationship with the quantity refers to the presence of high concentrations (greater than 5%) since they can have a negative impact on some parameters of animal productivity, such as the reduction in voluntary feed intake and on live weight gain or milk production [52].

Diets containing tannins reduce ammonia N in the rumen and the concentration of urea in milk. Therefore, less N is lost in the form of ammonia due to the reduction in the rumen protein degradation [53]. It is considered that moderate amounts, i.e., less than 5% of tannins in the diet offered to lactating cows can change the route of the N excreted, result in less excretion of urine and a greater excretion of feces, and this is achieved without affecting the efficiency of N utilization for milk production [54].

### 4.2. Herbage Intake and Digestibility

The values of the external marker content (chromium oxide) in the cattle feces indicated that C4 plants are predominant in the diet of the grazing cattle in both stargrass monoculture pastures and in the SPS. In this study, the digestibility of stargrass for the nitrogen fertilization range used was close to that shown by Johnson et al. [53] (54 to 61%), whereby lower digestibilities in the driest periods were noted. For Leucaena, the results seem high if we compare them with those found by Gonzales-García et al. [55] for the rainy (56.7%) and dry (52.6%) seasons. On the other hand, Barros-Rodríguez et al. [56] reported that the digestibility of Leucaena varies between 60 and 70% in vivo, indicating that the results here are within the range and the variations that are found in the literature; thus, this may be related to the age of the plant and the fractions of stems and leaves collected.

The similar intake between the systems coincides with the fact that the same stocking rate was used for all treatments. On the other hand, the amount consumed seem low, regardless of the pasture or the time of year; however, it may be related to the daily supply of concentrate, which fulfills the nutritional requirements of the animals. Leucaena intake was low compared to previous research. Valdivia [57] reported 2.6 and 2.3 kg of animal DM day^−1^ in creole cows in a silvopastoral system, which were implemented with and without energy supplementations, respectively. Bacab-Pérez and Solorio-Sánchez [27] found a higher intake of Leucaena in cows that remained for 20 h in two silvopastoral systems, with higher Leucaena densities and lower animal loads compared to the present experiment. These authors reported intake levels of 2.96 kg of DM in pastures with a density of 34,500 plants ha^−1^ and 3.0 AU ha^−1^, as well as 4.97 kg of DM in the pastures with 53,000 plants ha^−1^ and which were under 2.5 AU ha^−1^.

Despite Leucaena being an excellent forage, with great acceptability by cattle, imbalances in the protein:energy ratio consumed can lead to an excess of N in the diet. This affects the synthesis of microbial protein and, consequently, raises the ammonia levels in the blood, which can reduce voluntary intake and, subsequently, animal productivity [57]. Crude protein intake above the recommended requirements may lead to an increase in the energy requirements for the maintenance of cattle [58].

### 4.3. Excretory Pattern and N Balance

Urinary volume is influenced mainly by water intake, but also by the intake and excretion of mineral salts. Thus, large variations in the volume and frequency of urine are expected between animals, as well as throughout the day. On average, bovines urinate between 1.2 and 2.1 L per event, although lower volumes are more frequent for beef cattle. As the average frequency is 8 to 12 events per day [38], it can be said that common volumes of cattle urine would fluctuate between 9 to 24 L per day^−1^. The volumes found in this study are within this range. Fecal production is also quite variable and depends on the quantity and quality of the forage consumed; it is common for it to be between 11 and 16 defecations per day, which weigh between 1.5 and 2.7 kg of fresh feces per event [38]. Haynes and Williams [38] found an average of 15% dry matter in fresh feces; thus, the range of dry weight for daily defecation would be between 2.5 to 6.5 kg of dry feces day^−1^. In this case, the fecal production measured in this study was lower than the minimum expected for dairy cows, which may be related to the high amount of highly digestible concentrate offered to the animals, and also possibly to a slight protein/energy imbalance, which it would imply a lower efficiency of utilization in the N ingested [49].

### 4.4. Nitrous Oxide and Methane Emissions from Excreta

The dynamics of N in cattle systems is also an environmental concern since excess N in the system can exacerbate emissions of nitrous oxide (N_2_O), which is one of the most important greenhouse gases in livestock production. The return of N to the system through the urine of cattle implies higher emissions of N_2_O compared to the return that is obtained through the feces [22]. This is because it is a form of N that is rapidly available in the soil as mineral N, which is a substrate for the nitrification and denitrification processes in the soil that result in N_2_O emissions. In the present study, it was verified that the application of cattle urine to the calcareous soils of Yucatán in a monoculture of stargrass also highlighted more intense flows of N in the form of N_2_O when compared to that which originate in the feces (Figure 4).

At the beginning of the rainy season, the application of excreta to the soil resulted in one of the highest gas emission peaks, probably due to the relatively high rainfall of about 20 mm. Precipitation is an important factor in N_2_O production processes since denitrification is responsible for the reduction in oxidized forms of N to N_2_, and it is the most relevant process for gas emissions. A greater volume of rain increases the saturation of the pore space of the soil, which leads to the rapid consumption of existing O_2_ and stimulates the action of denitrifiers [9]. On the other hand, the temperature in the region remained at levels that were relatively favorable to denitrification, and the availability of the mineral N in the soil certainly increased with the excreta; these are all key factors for the emissions of N_2_O [59].

Enteric CH_4_ production is driven primarily by the level of feed intake and dietary fiber concentrations. Methane production increases with greater intakes due to the effects on ruminal passage rate and carbohydrate fermentation [20] The results of the accumulated emission of CH_4_ showed that the feces of the SPS systems resulted in a higher accumulation of gas in the rainy season (29.8 g C ha^−1^), which was followed by the feces of the MS system in the dry season (26.0 g C ha^−1^). The production of CH_4_ in the feces is dependent on the permanence of anaerobic conditions under low redox potential; therefore, after exposing the feces to the environment, the tendency is for the gas production to decrease with the loss of water in the feces. CH_4_ production occurs in a strictly anaerobic environment, and it has a positive correlation with soil moisture content [60]. In addition, increasing temperature raises methanogenesis rates, thus favoring an increase in the respiration rates and the consequent O_2_ depletion.

## 5. Conclusions

The use of Leucaena as a forage legume associated with stargrass in SPS systems with cattle provides a diet rich in CP, as was confirmed by the N consumed by the cows in an SPS when compared with the MS of stargrass only. However, milk production was similar in both systems, indicating that there was a slight nutritional constraint in converting Leucaena protein into milk. In any case, the presence of legumes allows a milk yield that is similar to that of an MS fertilized with 100 kg of N ha^−1^ year^−1^.

The diet which contained Leucaena (SPS) did not change the distribution of the N excreted through the urine and feces of cattle but instead reduced the emissions of N_2_O and CH_4_ from the excreta.

## Figures and Tables

**Figure 1 animals-13-01941-f001:**
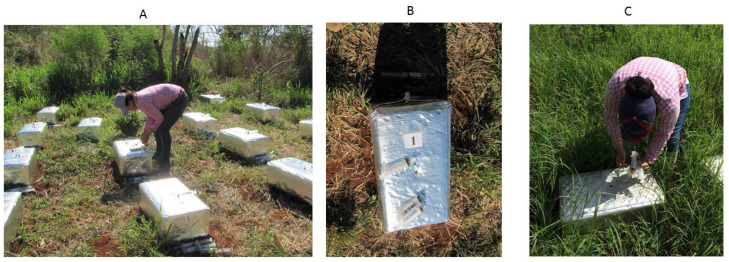
Static chamber technique for measuring the nitrous oxide fluxes from agricultural soils. The camera was checked to be working properly before taking measurements (**A**). The chamber was made mainly with stainless steel, aluminum peel/seal and polyethylene syringes (**B**). Gas sampling (**C**).

**Figure 2 animals-13-01941-f002:**
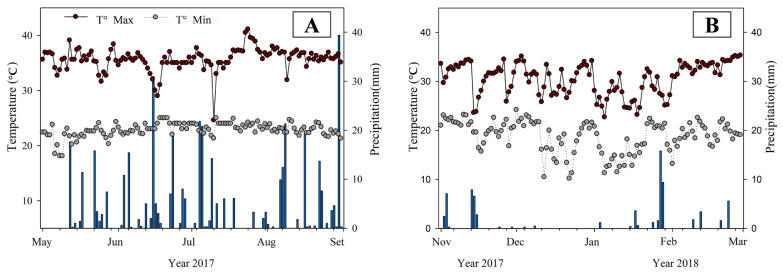
The maximum and minimum temperatures, and the extent of precipitation in the rainy (**A**) and the dry (**B**) seasons in the experimental area of Mérida, Yucatán, Mexico. °C = Temperature in degree Celsius.

**Figure 3 animals-13-01941-f003:**
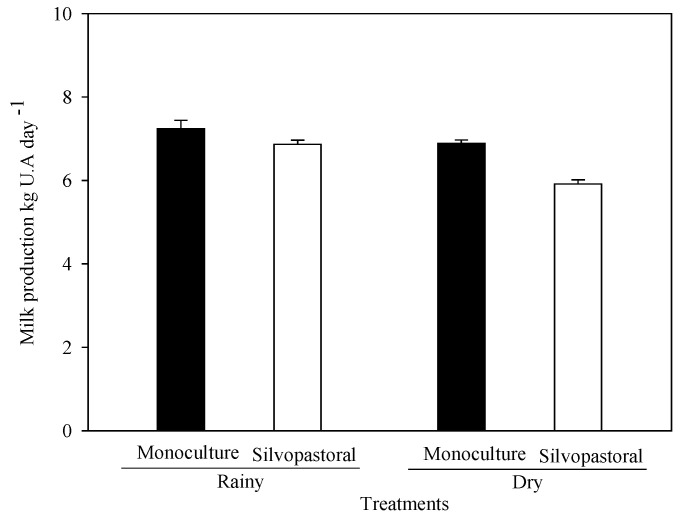
Milk production (kg AU day^−1^) of the cows in the MS and the SPS for the rainy and dry seasons. No statistical differences were found following conducting Tukey’s test (*p* = 0.05).

**Figure 4 animals-13-01941-f004:**
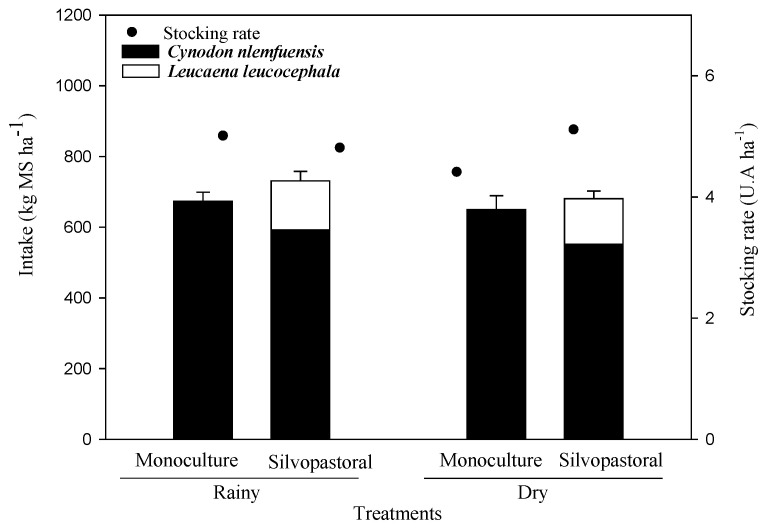
Intake (kg DM ha^−1^) of stargrass (MS and SPS) for the rainy and dry seasons (estimated by the chromium oxide technique). No significant differences were found between the treatments.

**Figure 5 animals-13-01941-f005:**
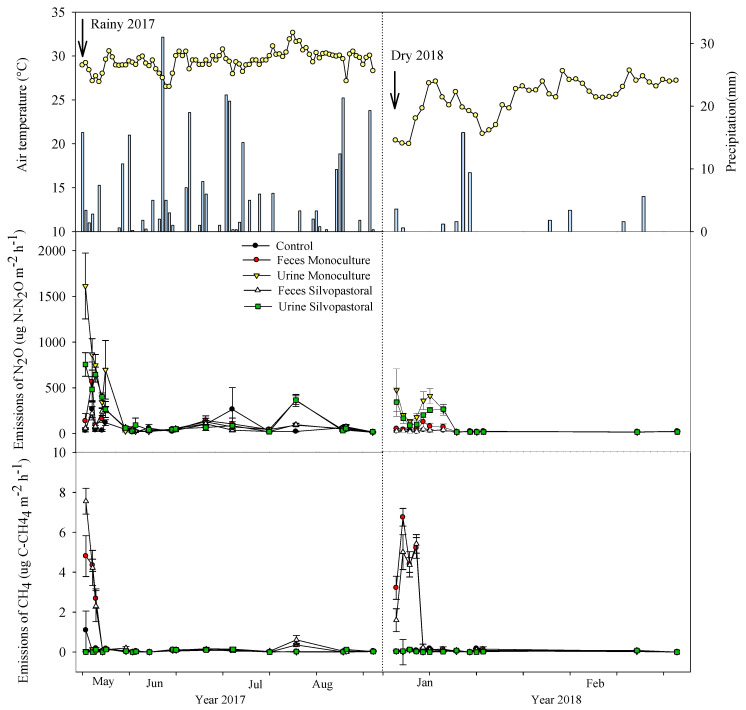
Precipitation (mm), mean temperature (°C), as well as the flows of N_2_O (µg N-N_2_O m^−2^ h^−^^1^) and CH_4_ (µg C-CH_4_ m^−2^ h^−^^1^) obtained from the soil under pasture of stargrass, which was treated with the excreta from animals that were in the MS and SPS.

**Table 1 animals-13-01941-t001:** The main physical and chemical characteristics of the soil of the experimental area.

Block	Soil–Rock Content %		pH	N	C	P	K	Ca	Mg
	Soil	Rocks		%		mg/kg			
I	22	78	7.8	0.89	6.4	28	530	872	352
II	40	60	7.8	0.98	5.0	45	565	824	328
III	21	79	7.9	0.99	7.2	81	457	1077	310
IV	19	79	7.9	0.96	6.1	111	517	1573	388
Median	26	74	7.8	0.95	6.2	66	517	1086	345

**Table 2 animals-13-01941-t002:** Proportion of Leucaena in the diet, and the urine and feces production of the cows that grazed in the MS and SPS for the rainy and dry seasons in Mérida, Yucatán, México.

Parameter	Rainy		Dry	
	MS	SPS	MS	SPS
% of *L. leucocephala* ^a^	0.0	19.0	0.0	18.9
Urine b (L/AU/day^−1^)	13.48 ± 1.06	11.34 ± 1.53	13.16 ± 1.39	13.21 ± 1.53
Feces c (kg DM/AU/day^−1^)	1.55 ± 0.06 aA	1.75 ± 0.06 bA	1.7 ± 0.10 B	1.53 ± 0.05 B

^a^—percentage of Leucaena material in the diet of the animals, as estimated by the ^13^C technique applied to the material of the feces of the animals combined with the digestibility information of the legume; b—estimated by “spot” sampling; and c—estimated by the chromium oxide external marker technique. Lower case letters represent a comparison of the means of the pasture type within the season, and upper-case letters are the comparison of the means between the seasons for the same pasture. Different letters indicate significant differences (according to the Tukey test (*p* = 0.05)).

**Table 3 animals-13-01941-t003:** The intake and excretion of N by cows present in the MS and SPS in the rainy and dry seasons in Yucatán, México.

Item	Rainy		Dry	
	MS	SPS	MS	SPS
	Nitrogen Balance (g AU/day^−1^)			
N intake	115.0 ± 4.3 bB	172.0 ± 6.3 *aA	144.9 ± 8.7 A	142.7 ± 4.5 B
N total exported	116.3 ± 4.4 B	137.6 ± 8.0	138.4 ±13.5 A	130.2 ± 11.1
N urine	46.8 ± 3.0 B	62.4 ± 12.3	69.3 ± 8.8 A	66.8 ± 5.3
N feces	32.5 ± 2.0 b	43.9 ± 2.5 aA	34.3 ± 2.4	34.4 ± 1.2 B
N milk	37.0 ± 1.7	31.3 ± 1.4	34.8 ± 3.3	29.0 ± 1.3
Balance	−1.3 ± 3.1 b	34.4 ± 4.3 aA	6.5 ± 13.6	12.5 ± 9.0B

The * indicates that differences were found between the amounts of N consumed, as well as the N that were excreted (according to the t-Student test (*p* = 0.05)). Lowercase letters represent a comparison of the system-type means within the season, and uppercase letters are the comparison of the means between seasons for the same system. Different letters indicate significant differences (according to the Tukey test (*p* = 0.05)).

**Table 4 animals-13-01941-t004:** The emission factor of the N_2_O and CH_4_ that arose from the bovine excreta originating from animals in the MS and SPS for the rainy and dry season in Mérida, Yucatan, Mexico.

Gas (EF Unit)	Rainy				Dry			
	MS		SPS		MS		SPS	
	Feces	Urine	Feces	Urine	Feces	Urine	Feces	Urine
N_2_O (%)	0.05	0.52 a	0.01	0.17 b	0.02	0.05	0.00	0.05
CH_4_ (kg CH_4_ AU^−1^ year^−1^)	0.17a	-	0.26 b	-	0.17 aA	-	0.14 a	-

EF means that each of the gases that are followed by the same lowercase letter were not different and the capital letters indicate a significant difference according to the Tukey test (*p* = 0.05).

## Data Availability

Not applicable.

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
