# Peer review of "Greenhouse Gas Emissions and Crossbred Cow Milk Production in a Silvopastoral System in Tropical Mexico"

_animals, 2023, doi:10.3390/ani13121941_

Round 1

Reviewer 1 Report

Please refer to my comments in the PDF

sometimes redundant use of terms.

Author Response

Dear reviewer, many thanks for your valuable comments and suggestions. Please, attached find the responses to your comments.

Kind regards

FJ

Reviewer 2 Report

The manuscript addresses production and quality of forage, milk production, and methane and nitrous oxide emissions from bovine feces and urine in two production systems: conventional grazing (grass in monoculture) and silvopastoral system (association of leguminous shrubs with grass). This topic is really very relevant. The article has scientific merits and most of the methods are adequate. Here are the considerations of the present evaluator:

ABSTRACT (lines 29-45): properly presented. It satisfactorily represents the study carried out.

INTRODUCTION (lines 48-104): satisfactorily addresses the relevance of the topic and contextualizes the problem for the reader.

MATERIAL AND METHODS (lines 105-293): the material and methods are mostly well written and adequate according to the experimental proposal. However, one issue worried the present evaluator.

Lines 158-164: This method is not accurate. Especially when forage accumulation is not considered (as apparently shown in the equation, it was not). I request further clarification if it really was not considered forage accumulation. If it really wasn't, in the opinion of the present evaluator this would be an experimental error with an impact on the estimate of consumption, and consequently, on other variables and results obtained. If this is the case, I believe that it would be appropriate to remove all variables that derive from dry matter consumption, and make a new submission.

RESULTS (lines 294-409): the results are presented in a very satisfactory and adequate manner.

DISCUSSION: (lines 410-551): The discussion is presented in a robust and adequate way, based on the presented results.

CONCLUSION (lines 552-560): compatible with the proposal of the manuscript and based on the observed results.

Author Response

Dear Review, many thanks for your important comments and suggestions. Please attached find the response to your comments and suggestions.

Many thanks,

kind regards

FJ

Reviewer 3 Report

General comments:

The study presented by Sarabia-Salgado et al. evaluates the impact of two grazing systems on production performance and climate-changing gas emissions in a tropical silvopastoral context.

The study is of some scientific and practical interest. Well conducted and reasonably well presented.

However, in my opinion, some changes would have to be made before the manuscript could be considered for publication.

In general, the paper is reasonably well structured. The introduction defines the problem adequately, the goal of the study and the experimental design are consistent.

The materials and methods section is well structured, clear and sufficiently complete, although it could be made more comprehensible and fluent.

The results are adequately reported and discussed.

specific comments:

L 37: reference is made to two sessions that have not been previously described;

L 38: AU, MS and SPS must be defined before using acronyms, check all text;

L 40 what is "urine treatment"?;

L 50-52: this statement should be supported with a bibliographic reference;

L 215: a parenthesis is missing;

L 239-243: it would be better to also include a photograph of the instrument in operating position;

L 284: if a mixed model was used, I expect that there is at least one random factor in the analysis, explain;

L 298: add "A" in (Figure 1);

Figure 1: add "A" and "B" in the graphs;

L 312: data that are not tabulated should be presented in the text as Mean±SD. Edit through the text.

Figure 2: if no statistical differences have been found, what is the meaning of the letters in the graph? perhaps it is better to eliminate them;

Figure 4: Graphs should be reported more legibly, perhaps separating MS and SPS.

In the discussion section, entire parts of the results section are reported.

Naturally, the results obtained can be recalled for the purposes of comparison with what was obtained by other authors, but not repeated exactly as in the lines: 463-468, 474-480, 514-518, 538-546;

Finally, always use the same indication (ie. star grass or stargrass, but not both; MO or MONO, but not both, and so on). Check all text

Author Response

Dear Review, many thanks for your important comments and suggestions. Please attached find the response to your comments and suggestions.

Many thanks

Kind regards

FJ
